# Female Infertility in Chronic Kidney Disease

**DOI:** 10.3390/diagnostics13203216

**Published:** 2023-10-15

**Authors:** Mahua Bhaduri, Ippokratis Sarris, Kate Bramham

**Affiliations:** 1King’s Fertility, Denmark Hill, London SE5 8BB, UK; 2King’s College, London WC2R 2LS, UK; kate.bramham@kcl.ac.uk

**Keywords:** chronic kidney disease, fertility, infertility, in vitro fertilization

## Abstract

This review summarises the current literature regarding infertility in women with chronic kidney disease (CKD), describing the epidemiology, pathophysiology, investigations, and management options. The pathophysiology is multifactorial, with proposed mechanisms including disruption of the hypothalamus−pituitary−ovarian axis, chronic inflammation, oxidative stress, psychological factors, and gonadotoxic effects of medications such as cyclophosphamide. Diagnostic investigations in CKD patients seeking to conceive should be considered earlier than in the healthy population. Investigations should include hormonal profiling, including markers such as Anti-Mullerian Hormone and imaging such as ultrasound, to evaluate ovarian reserve and identify gynaecology pathology. Treatment options for infertility in CKD patients include GnRH agonists to preserve ovarian function during cyclophosphamide treatment, as well as assisted reproductive technologies including in vitro fertilisation and ovulation induction. However, these treatments must be tailored to the individual’s health status, comorbidities, fertility requirements, and CKD stage. In conclusion, fertility is an important consideration for women with CKD, necessitating early investigation and tailored management. Early discussions regarding fertility are important in order to understand patients’ family planning and allow for prompt referral to fertility services. While challenges exist, ongoing research aims to clarify the underlying mechanism and optimise treatment strategies, which are crucial for improving quality of life and overall health outcomes.

## 1. Introduction

Chronic kidney disease (CKD) is defined by the ‘Kidney Disease: Improving Global Outcomes (KDIGO) CKD 2012 Guideline’ as either a reduction in kidney function below defined thresholds of the estimated glomerular filtration rate (eGFR—calculated from serum creatinine, age, and biological sex) for more than three months or people with structural kidney damage, and is associated with increased morbidity and mortality [1,2]. The prevalence of CKD in women of reproductive age is increasing, rising from 13.2% for the period 1999–2010; it is estimated to affect 16.7% of women by 2030 [3]. Concurrently, female infertility has increased, demonstrated by a recent World Health Organisation report which highlighted that 1 in 6 people experience infertility [4]. The increasing prevalence of CKD, combined with advances in medical treatments, has resulted in women living with CKD seeking advice concerning infertility more frequently, thus posing a new challenge for clinicians. In this review, we will explore the epidemiology, proposed pathophysiology, diagnostic investigations, and potential management options for women with infertility and CKD.

## 2. Epidemiology of Female Infertility in Chronic Kidney Disease

According to the most recent cross-sectional survey conducted by the National Health and Nutrition Examination survey (NHANES) in the USA, approximately 15% of patients with CKD are less than 50 years old, whereas a systematic review by Hall et al. estimated this proportion could be as high as 25.7% worldwide [5,6,7].

## 3. Definition of Infertility

Infertility is the disease of the male or female reproductive system defined by the failure to achieve pregnancy after 12 months or more of regular unprotected sexual intercourse [8]. Pregnancy rates in women with CKD are not known, as renal disease is not routinely screened for in pregnancy. However, using multiple national registries, it is predicted that among renal transplant recipients and dialysis patients, pregnancy rates are 10% and 1%, respectively, of the background age-adjusted rate [9].

## 4. Pathophysiology of Infertility in CKD Female Patients

The pathophysiology of how CKD impacts fertility is poorly understood, but there are several proposed mechanisms, and it is likely that more than one may be contributory.

### 4.1. Hypothalamus−Pituitary−Ovarian (HPO) Axis

The kidneys play an important role regulating the hypothalamus−pituitary−ovarian (HPO) axis. It is hypothesized that in women with CKD there is loss of the pulsatile release of Gonadotrophin releasing hormone (GnRH), which impairs follicle-stimulating hormone (FSH) and luteinizing hormone (LH) production and leads to low oestrogen and loss of ovulation. The loss of LH surge results in a lack of ovulation and subsequently oligomenorrhoea [9]. As eGFR declines, the loss of pulsatile GnRH release increases, resulting in patients experiencing oligomenorrhoea. This later leads to amenorrhoea, which affects 35–59% of women with kidney failure on dialysis [10]. However, the eGFR threshold at which this becomes clinically relevant is not known.

The impact of CKD on the HPO axis is based on a small study published over forty years ago on samples derived from 17 premenopausal women on haemodialysis with 15 normal controls [11]. Three to six blood samples were obtained from each subject and in seven women, daily blood samples were taken for 28–30 days. The bloods were tested for FSH, LH, prolactin, oestradiol, and progesterone. The plasma oestradiol, progesterone, and FSH levels were generally comparable to the controls during the follicular phase, but the oestradiol and LH levels did not reach the levels seen during normal midcycle peaks. Progesterone increased in only one of the patients in the luteal phrase. The lack of LH surge and absence of progesterone rise indicated that ovulation only occurred in one of the 17 patients. This patient had regular menses. Prolactin was found to be four-fold higher in the women with CKD compared with the healthy controls [11]. Prolactin is a hormone that is secreted in the anterior pituitary and stimulates milk production postpartum and inhibits ovulation. A high concentration of prolactin provides a negative feedback loop, reducing FSH and LH secretion and stopping ovulation. Therefore, elevated concentrations of prolactin coupled with the pulsatile loss of GnRH may contribute to the significantly decreased pregnancy rates in women living with CKD.

Previous research has focused on the impact on fertility in the late stages of CKD. It is proposed that the effect of CKD on menstruation and fertility is proportional to the degree of renal dysfunction; however, the relationship between less severe kidney disease and fertility has not been established [10,12].

### 4.2. Inflammation and Oxidative Stress

Patients with CKD have been shown to have high levels of inflammatory and oxidative stress markers [13]. Studies have reported that inflammatory and oxidative stress markers are increased during a single session of haemodialysis, mimicking changes that occur during acute immune activation. Infertility has also been associated with increased markers of low-grade inflammation. A Danish study of 2140 patients found that interleukin-6 (IL-6) concentrations were significantly higher among men with male factor infertility compared with the controls [14].

Females with primary ovarian failure have also been demonstrated to have an increased expression of interferon-γ and tumour necrosis factor alpha (TNF-α). These cytokines can induce major histocompatibility complex (MHC) class II antigens, which have been shown to be present in higher levels in ovarian tissue from these patients compared with the healthy controls [15]. It is thought that these cytokines could be contributing to an immune-mediated component of ovarian failure.

Moreover, patients with endometriosis, who are well known to have lower levels of fertility, have been shown to have higher concentrations of macrophages, as well as expressions of IL-1, IL-6, and TNF-α in their peritoneal fluid [16]. Pro-inflammatory cytokines are also thought to play an important role in implantation and placentation, although the precise mechanism is not fully understood [17].

It has been hypothesized that low-grade chronic inflammation associated with CKD could negatively impact fertility in women with kidney disease. However, the association between inflammation and infertility in women with CKD has not been well investigated and work is ongoing, using proteomics, to examine the role of these inflammatory markers further.

### 4.3. Cyclophosphamide

Cyclophosphamide is a medication commonly used to treat rapidly progressive glomerulonephritis and systemic lupus erythematosus, which are more prevalent in reproductive-aged patients. However, its gonadotoxcity effects are well established [18,19]. In females, it is known to cause ovarian impairment and failure. The effect on the ovaries is thought to be dose-dependent and age dependent, i.e., a higher cumulative dose and older age at the initiation of treatment increases the risk of cyclophosphamide-induced ovarian failure [18,20]. Ioannidis et al. demonstrated that women aged 32 or older had much higher rates of amennorrhoea compared with those who were younger [20]. This is because the ovarian follicular reserve reduces with age, so the gondotoxicity hit is greater in older women. It is often thought that if the dose of cyclophosphamide is ‘low’ it does not affect the ovaries. This is supported by findings from the Euro-Lupus study regimen, which reported no significant difference in ovarian reserve (determined by Anti-Mullerian Hormone titres) before and after low-dose intravenous cyclophosphamide treatments [21]. However, this sub-analysis was limited by a small sample size (*n* = 10). In addition, those receiving high-dose IV cyclophosphamide (≥6 g) did show a significant decline in Anti-Mullerian Hormone (AMH) titres [21], while women with repeated cycles of cyclophosphamide were excluded. Therefore, the exact threshold dose remains unknown.

For reproductive-aged females who wish to conserve their fertility there are therapeutic options available, including oocyte cryopreservation via ovarian stimulation or cryopreservation of ovarian tissue. These will be covered later in this review. However, patients are often too unwell to undergo these treatments or do not have enough time prior to starting cyclophosphamide, as this often requires urgent commencement. Moreover, fertility-sparing treatments are frequently needed urgently, and are not well understood by nephrologists and thus are not discussed until later in the patient’s journey [22].

### 4.4. Sexual Dysfunction in CKD

Reduced libido, difficulty with arousal, dyspareunia, and anorgasmia are thought to be common in pre-dialysis female patients with CKD, with low levels of oestrogen leading to reduced vaginal lubrication, resulting in pain during sexual intercourse [23,24]. A study of 52 females with kidney failure used the validated Female Sexual Function Index (FSFI) and Beck Depression Inventory (BDI) and found they had lower sexual function scores, with pain and reduced lubrication, compared with the healthy age-matched controls [25,26]. CKD, as well as its co-morbidities and the treatments used, can have a negative impact on sexual function. Although the burden of sexual dysfunction is difficult to quantify in this cohort, it appears that there is a positive correlation with disease progression and incidence of sexual dysfunction. Sexual desire has been found to be reduced in half of dialysis patients [27]. Even after transplants, sexual dysfunction is found to be more common in transplant patients compared the healthy volunteers [28]. Medications commonly used to treat people with CKD, including diuretics, beta-blockers, and/or steroids, may all negatively impact sexual functions [29]. Moreover, high rates of negative body image and depression have also been described in women with CKD [25,27,30]. The paucity of data in this field could be owing to a lack of patient drive and voice in this area due to stigma around the subject, but a recent Canadian review highlighted the desire of patients for it to be an important research priority [31].

### 4.5. Reproductive Choice

The extent to which low fertility rates in women with CKD is related to reproductive choice versus impaired fertility remains unclear. A survey of Dutch nephrologists reported that 88% advised their female patients not to become pregnant [22]. A systematic review of qualitative studies in people with CKD reported that women often expressed ‘an innate desire’ to have children, but feared it would impact their health and felt traumatized when discouraged by their doctors [32]. The review highlighted that further insight was needed to explore the perceptions of women with CKD, focussing on their renal disease, how it impacts their reproductive potential and their views on assisted reproductive technology (ART).

## 5. Diagnostic Testing and Investigations in a Patient with CKD and Infertility

### 5.1. History Taking

It has been suggested that patients with CKD should be investigated for infertility earlier than that of the healthy population. Wiles et al. suggested that a female of reproductive age with CKD should be referred to a specialist fertility centre after six months of trying to conceive rather than one year, as advised in the healthy cohort [9]. As with any disease, careful history taking is an important tool to diagnose infertility as well as to investigate the potential causes (see Table 1). In a female patient with CKD, often the cause can be due to anovulation secondary to the dysfunction of the HPO axis and hyperprolactinemia [9]. In this case, it is important to obtain a detailed menstrual history. Patients who have had peritoneal dialysis or renal transplant surgery have a greater chance of abdominal adhesions and thus potential fallopian tubal damage. Tubal causes of infertility (25%) were identified as the most common cause of infertility in women with CKD (*n* = 76) in a recent systematic review [33]. Tubal factor is thought to account for 11–67% of infertility diagnoses in the general infertile population [34,35,36].

It is important to ask the patient about current and previous medications, including doses, as there is often a dose-dependent relationship between gonadotoxic medications and fertility [18]. Consideration of teratogenic medications remains equally important. For example, most angiotensin-converting enzyme inhibitors (ACEi) or mycophenolate mofetil (an antiproliferative immunosuppressant commonly used in people with renal transplants) (MMF) are teratogenic and should be substituted prior to pregnancy or upon confirmation of pregnancy [37].

Patients with genetic conditions that cause kidney failure, such as autosomal dominant polycystic kidney disease (ADPKD), may wish to have pre-implantation genetic testing (PGT) to ensure their offspring are unaffected. In these cases, they should be informed of the fertility treatments available to them early in the disease process and referred to specialist centres when they show an interest in conceiving.

### 5.2. Investigations: Blood Tests

A hormonal profile can be important to ascertain the cause of infertility. For example, a raised FSH (over 30 IU/L) together with raised LH and low serum oestradiol levels indicates hypergonadotropic, hypoestrogenic anovulation caused by ovarian failure. If the patient is less than 45 years old, these results suggest primary ovarian insufficiency as the cause of infertility. High prolactin concentrations, as discussed previously, are commonly seen in CKD due to reduced renal clearance and increased secretion [38].

AMH is a glycoprotein, which has a role in folliculogenesis. It is synthesised in the granulosa cells of the ovaries. It is probably the most studied reproductive hormone in women with CKD because it is thought to be the most stable biomarker of ovarian reserve [39,40]. It is not hormone dependent and remains at a constant level during the entire menstrual cycle. Thus, the serum AMH concentration reflects the follicular pool and is considered the best biomarker of ovarian reserve. In the healthy population, AMH can be used to predict the response to ovarian stimulation in IVF in terms of oocyte yield, but it remains unclear whether AMH can be used to predict pregnancy or live birth in spontaneous or assisted conception [39,41]. AMH positively correlates with the antral follicle count (AFC) and decreases with age at a rate of 6% per year in the general population [42,43,44,45,46].

Wiles et al. analysed AMH according to the stage of CKD [47]. They recruited patients across the spectrum of CKD severity as part of the Pregnancy Adaptations In Renal disease Study (PAIRS) and reported that AMH concentrations were lower in all CKD stages compared with women without CKD. Stoumpos et al. reported AMH levels were 43% lower in women with renal failure (*n* = 27) compared with 600 age-matched controls [12]. Interestingly, they also found that AMH levels were slightly higher in patients having haemodialysis, and the mechanism for this finding remains unexplained.

Menses is frequently restored within a few months of successful kidney transplantation; hence, it could be postulated that post-transplantation AMH titres would return to similar concentrations as the general population. However, a study of 60 women showed that serum AMH concentrations remained reduced in patients who had a renal transplantation [48]. The authors suggest that immunosuppressive therapy used after renal transplantation could reduce AMH by damaging the granulosa cells. However, a study in patients with systemic lupus erythematosus who received immunosuppression did not show any difference in AMH concentrations [49].

It remains difficult to draw firm conclusions regarding the clinical role of AMH concentrations in women with CKD patients as studies have tended to have had small samples sizes and concentrated on the latter stages of CKD; hence, more research needs to be conducted to identify trends.

### 5.3. Investigations: Imaging

Ultrasound is the first line imaging modality in the assessment of female fertility. To date, there is only a single prospective study published in 1997 investigating the prevalence of gynaecological disorders in 100 women with CKD using pelvic ultrasound as part of their assessment. The authors reported that 58% of the cohort had menstrual disorders [50]. Ovarian reserve, which is a marker of fertility, can be evaluated on ultrasound. This is indicated by the size and morphology of the ovaries, as well as the antral follicle count (AFC). The antral follicle count is calculated by counting the number of follicles (measuring between 2–10 mm) seen within the ovaries upon ultrasound, with the total number of follicles seen in both ovaries being the AFC.

AMH and AFC are the best predictors of ovarian response to ART [51,52]. They both reflect the number of small antral follicles and are strongly correlated [40]. They are often used interchangeably in clinical medicine [39]. However, a systematic review reported that AMH had superior accuracy compared with AFC, likely due to the difficulties in the standardization of AFC determination due to intra- and inter- observer variability [39,51,53]. The same review also reported that AMH can be affected by co-morbidities including type 1 diabetes mellitus, acute onset Crohn’s disease, and haematological or other childhood cancers [54,55]. To our knowledge, there have been no studies correlating AFC and AMH to measure ovarian reserve in CKD patients.

Pathologies such as endometriosis, adenomyosis, and fibroids can be diagnosed upon transvaginal ultrasound. This helps with further assessing the causes of infertility and subsequent management. If surgical management is required, Magnetic Resonance Imaging (MRI) can be useful in surgical planning, particularly if fibroids or endometriosis are present. A summary of all the investigations which can be useful in diagnosing infertility and identifying the cause is demonstrated in Table 2.

## 6. Treatment for Infertility in CKD Patients

The management of infertility is primarily dependent on the cause of infertility and is complex and beyond the scope of this review. The main treatment options available, and what considerations are required in patients with CKD, are outlined.

### 6.1. GnRH Agonists

Cyclophosphamide, as described previously, can cause ovarian damage. To counteract this, GnRH agonists (GnRHa) can be used to preserve ovarian function and subsequently avoid infertility [56]. They induce a functional loss of gonadotropin stimulation and suppress the HPO axis [57,58,59]. A study of 40 women with lupus nephritis showed that patients who had leuprolide, a synthetic GnRHa, had lower rates of primary ovarian insufficiency [60]. The drug is usually administered intramuscularly or subcutaneously. The exact regime is dependent on the specific drug formulation used and clinical practice, but is usually continued throughout the cyclophosphamide treatment. GnRHa are more commonly used in the context of cyclophosphamide chemotherapy regimens, as the cyclophosphamide doses are greater, but there is a drive among the nephrology community to make it more commonplace in reproductive-aged female patients who undergo cyclophosphamide treatment.

### 6.2. Assisted Reproductive Technologies (ART)

ARTs include ovulation induction (OI), intrauterine insemination (IUI), egg freezing, in vitro fertilization (IVF), and intracytoplasmic sperm injection (ICSI). OI involves the use of either oral agents or injectable gonadotrophins in order to promote the growth of antral follicles to ovulatory ones. Oral agents include selective oestrogen-receptor modulators and aromatase inhibitors, such as clomiphene or letrozole, which increase the endogenous production of follicle-stimulating hormone (FSH) from the pituitary. In contrast, injectable gonadotrophins bypass the pituitary and act directly on the ovary and its follicles. These medications are typically given in the first part of the cycle to stimulate one or two dominant follicles that will ovulate [36]. OI can be used in conjunction with IUI, where sperm is prepared and injected into the uterus at the optimal time for fertilisation. OI is most often used in patients with polycystic ovarian syndrome (PCOS) or hypogonadotropic hypogonadal anovulation. There are no reported adverse incidents or side effects from these medications in patients with CKD [33].

IVF is where high concentrations of FSH-containing gonadotrophins are injected into the patient to stimulate the ovaries to produce more than one mature oocyte. The oocytes are collected from the ovaries via a needle passed through the vaginal wall and are fertilised with sperm in vitro [36,51,61]. Common risks of the procedure include post-procedure pain and minor bleeding, with much rarer complications including damage to surrounding structures, severe bleeding or collections, and infection [62]. In renal transplants, the kidneys are usually situated higher than the ovaries, so the risk of damage is rare [33].

Although fertility treatment is generally very safe, there are risks and potential complications. Ovarian hyperstimulation syndrome (OHSS) is a potential serious complication of fertility treatment caused by an over response to gonadotrophin intake. A systematic review of women with CKD undergoing IVF reported that 7.4% of women developed OHSS, of which AKI occurred in three cases (out of 54). This is higher than the incidence in the general population undergoing IVF, which is 3–6% [63], but there is likely reporting bias. Live birth rates and miscarriage were similar for women having ART with or without CKD, but more women with CKD developed pre-eclampsia, had babies born at lower gestational ages, and had lower birth weights [33]. The authors concluded that women with CKD could have successful fertility treatment with good pregnancy success rates and safety profiles, but more research was needed to understand the optimal management of infertility in these women [33].

## 7. Conclusions

It is well recognised that females with CKD can have a strong desire to have children, but they can also feel that their fertility status is being overlooked in their management [32]. The proposed mechanisms that underpin the association of CKD and infertility are not fully understood. Further work to better describe these processes is urgently warranted to identify potential new targets of therapy. Early referral to investigate fertility is encouraged, especially in individuals who are considered to be at high risk for infertility. The management of infertility in these females can be difficult due to their ongoing health conditions; hence, careful patient selection is particularly important in these cases.

## Figures and Tables

**Table 1 diagnostics-13-03216-t001:** History taking for a CKD patient suffering with infertility.

History	
Menstrual history	Duration and frequency of period, dysmennorrhoea, menorrhagia, intermenstrual bleeding, post coital bleeding, dyspareunia, vaginal discharge, and urinary and bowel symptoms
Previous gynaecology history	Contraception history, previous sexual transmitted disease, previous gynaecological surgeries, and known fibroids or ovarian cysts
Obstetric history	Gravidity, parity, mode of delivery, and previous pregnancy complications
Past medical history	Cause of CKD, associated comorbidities i.e., hypertension, diabetes, and suitability for pregnancy
Drug history	Immunosuppressants: avoid MMF, change to another agent at least 3 months before conceiving ACEI, ARB, SGLT-2—stop at conception Cyclophosphamide
Family history	Genetic disease or familial associations i.e., ADPKD and diabetes
Social history	Smoking, alcohol, and drug intake. Social support and performance status

**Table 2 diagnostics-13-03216-t002:** Summary of investigations used in the diagnosis of infertility.

Investigations	Parameters
Blood tests	Follicle stimulating hormone, luteinizing hormone, prolactin, testosterone, oestradiol, progesterone, Anti-Mullerian Hormone, b-HCG, thyroid function tests
Imaging modalities	Transvaginal and transabdominal ultrasound, Magnetic resonance imaging

## Data Availability

No new data were created from this paper.

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
