# Peer review of "Female Infertility in Chronic Kidney Disease"

_diagnostics, 2023, doi:10.3390/diagnostics13203216_

Round 1
Reviewer 1 Report
congratulations for this interesting review
it has been mentioned (line121-123):
The effect on the ovary is thought to be dose-dependent and also age dependent; i.e. a higher cumulative dose and older age at initiation of treatment increases the risk of cyclophosphamide-induced ovarian failure
older age should be changed to younger age at initiation !!
Author Response
Thank you for your comments and thorough review of our work.
The effect of cyclophosphamide on the ovary is dose and age dependant. The older you are, the more likely it will cause damage to the ovary. This is because as you get older, your ovarian follicular reserve reduces. I have tried to explain more thoroughly in the review-
'The effect on the ovary is thought to be dose-dependent and also age dependent, i.e. a higher cumulative dose and older age at initiation of treatment increases the risk of cyclophosphamide-induced ovarian failure [18,20]. Ioannidis et al demonstrated that women aged 32 or older had much higher rates of amennorrhoea compared to those who were younger [20]. This is because ovarian follicular reserve reduces with age, so the gondotoxicity hit is greater in older women.'
I hope this addresses your comment appropriately.
Reviewer 2 Report
Good overall review
Author Response
Thank you for your comment and we are glad you enjoyed reading our work.
Reviewer 3 Report
Dear Authors,
The presented study tackles an issue of female infertility in chronic kidney disease. I have read the article with a great interest. Overall, I think that this narrative review is more suitable for book chapter then an article. Meta-analyses following to the PRISMA guidelines would be more suitable in this case.
Moreover, I suggest following guidelines for authors from journal’s webpage (https://www.mdpi.com/journal/diagnostics/instructions)
e.g.
1. I suggest shortening the Abstract (max 200 words) -maybe by removing definition and prevalence.
2. I suggest separating definition from prevalence (in separate paragraphs)
3. Reference numbers should be placed in square brackets [ ], and placed before the punctuation; for example [1], [1–3] or [1,3].
4. The table should be prepared according to journal’s templet
Author Response
The presented study tackles an issue of female infertility in chronic kidney disease. I have read the article with a great interest. Overall, I think that this narrative review is more suitable for book chapter then an article. Meta-analyses following to the PRISMA guidelines would be more suitable in this case.
Thank you, we are pleased that you found this article interesting. We were invited and accepted to write a narrative review on female infertility in CKD, which we believe will sit well in this themed issue.
I suggest shortening the Abstract (max 200 words) -maybe by removing definition and prevalence.
I have shortened the abstract as you have suggested, removing the definitions and the prevalence. It is now 197 words.
I suggest separating definition from prevalence (in separate paragraphs)
Thank you for this suggestion. I have done this.
Reference numbers should be placed in square brackets [ ], and placed before the punctuation; for example [1], [1–3] or [1,3].
I have amended this.
The table should be prepared according to journal’s templet
I have prepared the table in the same format as the journal template.
Thank you for your thorough review and suggestions.
Round 2
Reviewer 3 Report
I have no other comments.